

**Diverse Causes of Extreme Rainfall in November 2023 over Equatorial**
**Africa**
**Hermann N. Nana[1]\* · Masilin Gudoshava[2] · Roméo S. Tanessong[3,1] · Alain T. Tamoffo[4] ·**
**Derbetini A. Vondou[1]**
[1]Laboratory for Environmental Modelling and Atmospheric Physics (LEMAP), Physics
Department, University of Yaounde 1, PO Box 812, Yaounde (Cameroon)
[2]IGAD Climate Prediction and Applications Centre (ICPAC), Nairobi, (Kenya)
[3]Department of Meteorology and Climatology; Higher Institute of Agriculture, Forestry,
Water and Environment; University of Ebolowa, PO Box 118, Ebolowa (Cameroon)
[4]Climate Service Center Germany (GERICS), Helmholtz-Zentrum Hereon, Fischertwiete 1,
20095 Hamburg, Germany
* Corresponding author: Hermann N. Nana (nanahermann100@yahoo.com)
Hermann N. Nana's orcid: https://orcid.org/0000-0002-0973-8613
Masilin Gudoshava's orcid: https://orcid.org/0000-0003-0315-9271
Roméo S. Tanessong's orcid: https://orcid.org/0000-0003-3804-5901
Alain T. Tamoffo's orcid: https://orcid.org/0000-0001-8482-8881
Derbetini A. Vondou's orcid: https://orcid.org/0000-0002-8681-5328



**Abstract**
Understanding the atmospheric factors that lead to extreme rainfall events is
essential to improve climate forecasting. This study aims to diagnose the physical
processes underlying the extreme rainfall event of November 2023 in Equatorial Africa
(EA), using the ERA5 reanalysis dataset. Composite, spatio-temporal and correlation
analyses are used to shed light on the relationship between the November 2023 extreme
precipitation events and the various associated factors. The analysis reveals that these
extreme rainfall were mainly controlled by several factors that occurred during this period
in the Pacific, Atlantic and Indian oceans. These factors include strong Sea-Surface-
Temperature (SST) anomalies in Niño-3.4, North Tropical Atlantic, Equatorial Atlantic and
Indian Ocean Dipole (IOD) oceanic regions, changes in zonal winds, the Walker circulation,
the anomalous moisture flux and its divergence, the easterly jets and the activity of the
Madden-Julian Oscillation (MJO). This convergence of moisture flows entered the EA
region through its western and eastern boundaries, coming from the equatorial Atlantic
and Indian oceans respectively. The juxtaposition of these factors has led to strong and
positive rainfall anomalies in EA, with the highest values over the East African region,
mainly over southern Ethiopia, Somalia, Kenya and Tanzania, which received more than
430 mm of rainfall during this month. Our findings suggest that many dynamic
atmospheric effects need to be taken into account jointly to anticipate this type of
extreme event. The results of the present study contribute to the improvement of sub-
seasonal to seasonal rainfall forecasts by the region's national meteorological services, to
enable us to increase the resilience of the region's citizens to these extreme weather
conditions.
**Keywords:** Equatorial Africa, IOD, atmospheric circulation, SST, rainfall variability










## 1. Introduction

In recent decades, Equatorial Africa (EA) has experienced an increase in the frequency and intensity of extreme events, particularly droughts, torrential rains and floods (Kilavi et al. 2018). In addition, climate-sensitive sectors such as water, transport, health and agriculture, among others, are negatively impacted by these events, which have recently increased in magnitude and frequency. Flooding from these extreme events leads to infrastructural and socio-economic damage, water shortages, severe human damage and socio-economic disruption (Funk et al. 2008; Tanessong et al. 2017). With the increase in greenhouse gases, the impacts of these extreme events continue and are projected to increase (Gudoshava et al. 2020; Ngavom et al. 2024). East and Central African countries are the regions influenced by high levels of intra-seasonal to inter-annual variability in monsoon rainfall (Lüdecke et al. 2021), which are the main flood-prone countries in Africa (Li et al. 2016). These exceptional flooding events generally occur during October and November months, which correspond to rainy months in Central and East Africa (Wainwright et al. 2020; Nicholson et al. 2022; Kenfack et al. 2024).

During November 2023, EA experienced a very wet period during which many parts of the region were affected by extreme rainfall events, most pronounced over East Africa where heavy rainfall and floods caused damage in several countries such as Somalia, Ethiopia, Kenya, Burundi and Malawi (https://floodlist.com/africa/east-africa-floods-november-2023-somalia-ethiopia-kenya-burundi-malawi), causing up to 90 deaths and more than 113,690 temporarily displaced. In Kenya, many areas were devastated by significant flooding. At least 19 of the country's 47 counties were severely affected by the floods, which started at the end of October 2023. More than 46 people lost their lives and over 58,000 people have been displaced by the increased heavy precipitation and subsequent flooding (https://floodlist.com/africa/kenya-floods-update-november-2023). In Tanzania, some 10,090 people, or 2,018 households, were affected, and 1,245 houses were damaged, with over 40 deaths recorded (https://floodlist.com/africa/tanzania-floods-landslides-hanang-district-december-2023). Extreme rainfall events also occurred in western EA regions. Democratic Republic of the Congo (DRC), Central African Republic (CAR) and Nigeria countries also experienced significant flooding and landslides which affected over 90,000 people, and around ten schools and health centres were destroyed (https://www.unocha.org/publications/report/burkina-faso/west-and-central-africa-weekly-humanitarian-snapshot-15-21-november-2023). These conditions have placed EA in a severe food crisis. Given that climate models project a trend of increased extreme rainfall in the region (Fotso-Ngeumo et al. 2019), and that the impacts of these extreme events are expected to increase (Gudoshava et al. 2020), it is therefore essential to understand the processes behind these extreme events.

Numerous studies have examined the different causes of November's extreme rainfall in the EA. They have shown that these extreme events were associated with numerous mechanisms linked to Sea-Surface-Temperature (SST) patterns in the tropical Pacific, Atlantic and Indian Oceans (Nana et al. 2023, 2024; Palmer et al. 2023; Roy and Troccoli 2024). These large-scale ocean drivers are the El Nino-Southern Oscillation (ENSO; Palmer et al. 2023), Indian Ocean Dipole (IOD; Nicholson 2015; Roy and Troccoli 2024),



North Tropical Atlantic (NTA; McHugh and Rogers 2001; Ingeri et al. 2024) and the
Equatorial Atlantic (ATL; Dezfuli and Nicholson 2013). Nicholson (2015) showed that
increased rainfall over East Africa is due to the presence of IOD in the October-December
season (OND). Following Wahiduzzaman and Luo (2020), several IOD episodes coincide
with an ENSO event, and Zhang et al. (2015) found that an ENSO episode can lead to the
development of an IOD event through the Walker circulation that connects the Indian and
Pacific Oceans. In this line, Roy and Troccoli (2024) have shown that the increase in rainfall
over East Africa is linked to the simultaneous presence of two factors, the IOD and ENSO.
Moihamette et al. (2022) conducted a similar study but focused on the months of
September-October-November over Central Africa. They found that during this period, the
warm (positive IOD with El Niño) and cold (negative IOD with La Niña) phases of the IOD
and ENSO frequently coincide. This study also showed that positive IOD events contribute
significantly to more rainfall in Central Africa after the El Niño effect is removed. Another
driver of East African rainfall is the NTA, which conducts more rainfall over many
countries, mainly Tanzania, Kenya and Uganda (Ingeri et al. 2024). Over western EA, the
Indian Ocean influences the climate system of this region through the ATL region, mainly
in October and November (Moihamette et al. 2024). Furthermore, in November, ENSO and
IOD were not considered to be important factors in many flood-affected regions,
particularly northern regions (north of 5° N; e.g. Moihamette et al. 2022). Consequently,
all the events of November 2023 were probably the result of the simultaneous occurrence
of several factors. These include SST in the Indian, Atlantic and Pacific oceans, the
atmospheric zonal circulation, Walker circulation, Madden-Julian Oscillation (MJO; Madden
and Julian 1971, 1972) activity, moisture transport and divergence and African Jets.
This study aims to identify and analyse the different factors that can sustain these
extreme rainfall events in Equatorial Africa. This paper is structured as follows: Data and
metrics used to diagnose mechanisms are described in Section 2 and features in the EA
rainfall and ocean SSTs are presented in Section 3. Physical processes and underlying
mechanisms associated with rainfall extremes are shown in Section 4. Section 5
summarizes and concludes the document.
**2. Data and Methods**
**2.1. Data**
ERA5 produces monthly estimates of climate variables on a global scale, featuring
a horizontal resolution of 31 km (0.25° x 0.25°) and 137 vertical levels ranging from the
surface to 1 hPa (Hersbach et al. 2020) available from 1979 through the present. In this
study, we extracted ERA5 monthly data for rainfall (precip in mm day$^{-1}$), SST (sst in K),
zonal and meridional winds (u and v in m s$^{-1}$), specific humidity (q in Kg Kg$^{-1}$), vertical
velocity (w in m s$^{-1}$), surface pressure (sp in Pa), total column water vapor (tcwv in mm) , 2-
meter dew point temperature (d2m in K), surface net solar radiation (ssr in J m$^{-2}$) and low
cloud cover (lcc in %). The data span 23 vertical levels, from 1000 to 200 hPa, and cover the
period from November 1981 to 2023. To assess the ability of ERA5 to reproduce rainfall
extremes that occurred in November 2023, the observational dataset from the Climate
Hazards Group InfraRed Precipitation with Station dataset (CHIRPS; Funk et al. 2015) is
used. This dataset includes high-resolution satellite imagery and station rain-gauge data,





available from 1981 through the present and has a high spatial resolution of 0.05° × 0.05°.
The SST dataset used in this paper to analyse the oceanic conditions is provided by the
Extended Reconstructed Sea Surface Temperature Version 5 (ERSSTv5; Huang et al. 2017).
The dataset is available from 1854 through present at a resolution of 2.0° × 2.0°.
**2.2. Methods**
The atmospheric factors explored in this study include the DMI, zonal winds, the
Walker circulation over EA and the Oceans, moisture flux and divergence fields, tropical
SSTs in the Pacific, Atlantic and Indian Oceans, and tropical waves, namely African Easterly
Waves and the Madden-Julian Oscillation (MJO). We are focusing on these factors because
they represent the main contributors to extreme rainfall events in the EA (Kuete et al.,
2019; Nicholson et al. 2022; Roy and Troccoli 2024; Gudoshava et al., 2024). The DMI is
calculated as the difference between SST anomalies in a western (60° E-80° E, 10° S-10° N)
and an eastern sector (90° E-110° E, 10° S-0° S) of the central Indian Ocean. Since several
oceans were anomalous during these extreme events, oceanic conditions of more regions
have been analysed. These are: the North Tropical Atlantic (NTA; 5°-15° N and 40°-15° W),
the Equatorial Atlantic (ATL; 3° S-3° N and 20° W-10° E) and  the Niño-3.4 (5° S-5° N and
170°-120° W).
We start our analyses by characterizing rainfall distribution as shown by both
CHIRPS and ERA5 over EA (defined as 10° S-10° N; 10°-50° E, see red box in Figure 1).
Afterwards, processes associated with November anomaly patterns are diagnosed. We
first look at November anomalies in SSTs and specific humidity on the one hand, and then
anomalies in wind and moisture flux, on the other. The zonal and meridional circulation
can be modulated by variations in winds and specific humidity, which can have an impact
on the regional hydrological cycle, either by enhancing or weakening it, following the
findings by Seager et al. (2010) and Tamoffo et al. (2024).
We have also investigated the water vapor mass transported within the zonal
circulation by estimating the zonal mass-weighted stream-functions ($\Psi_Z$; Stachnik and
Schumacher 2011; Taguela et al. 2022), following the Equation 1:
$$\Psi_Z(p) = \frac{2\pi R}{g} \int_{sp}^{P\,top} [u]\, dp \qquad (1)$$
where R is the Earth's radius (m), g is the constant of gravity, *sp* and *P top* the surface and
top-level pressure respectively, and bracket terms denote the meridionally averaged of
the zonal wind over the latitudes 10° S-10° N.
Note that the CB cell is characterized by the negative values of the zonal mass-
weighted stream-function ($\Psi_Z < 0$). This function is used to characterise the Walker-type
circulation over the Western EA (Longandjo and Rouault 2020; Tamoffo et al. 2022).
Environmental conditions for November 2023 are also analysed through an assessment of
vertically integrated moisture transport. Vertically integrated moisture flux (Q; Zheng and
Eltahir 1998) is calculated using Equation 2 given as follows:



$$Q = \frac{1}{g} \int\limits_{sp}^{P\,top} qV \; \mathrm{dp} \qquad\qquad (2)$$

where V is the horizontal wind (m s$^{-1}$).
For all metrics used in this study, composite anomalies are obtained by removing
the 42-year average of the period 1981-2022. For significance testing, the student's t-test
is applied.

## 3. Rainfall in November 2023

Figure 1 shows the percentage contribution of November rainfall to total annual
rainfall (Fig. 1a,d), November mean rainfall (Fig. 1b,e), and the difference between
November 2023 and the long-term Mean (LTM) rainfall (1981-2022; Fig. 1c,f). In general,
November 2023 was exceptionally wet throughout the EA, with more rainfall over the
eastern Africa area, where monthly anomalies were typically up to 7 mm day$^{-1}$ than that
occurring over the western area (Fig. 1c,f). Fig. 4a features Indices of standardised rainfall
anomalies since 1981, based on both CHIRPS and ERA5 dataset. Extreme positive rainfall
anomalies occurred in 1982, 1994, 1997, 2006 and 2019 in most parts of equatorial Africa.
November 2023 is the strongest since 1981.
Positive anomalies prevail from South Sudan and Ethiopia, around 10 to 15° N, to
at least 10° S over the eastern region, and over northern (equator to 15° N) and southern
(15° S to 5° S) regions, over the western EA area in CHIRPS data (Fig. 1c). There is generally
good agreement between ERA5 and CHIRPS except in Congo Republic, northern part of
Gabon and central part of DRC, where ERA5 does not well estimate the negative rainfall
anomalies (Fig. 1f), and in eastern Africa regions, where ERA5 shows weaker rainfall than
CHIRPS. During this month, most parts of the EA region received increased rainfall of up
to 2 mm day$^{-1}$. These areas (except the northern regions) coincide with those that strongly
contribute to the annual EA rainfall (Fig. 1a,d) and normally receive at least 3 mm day$^{-1}$ of
total rainfall (Fig. 1b,e). In the typically dry northern regions during November, rainfall
ranged between 3 and 9 mm day$^{-1}$, exceeding the long-term mean (LTM) by approximately
2 to 8 mm day$^{-1.}$

6    6





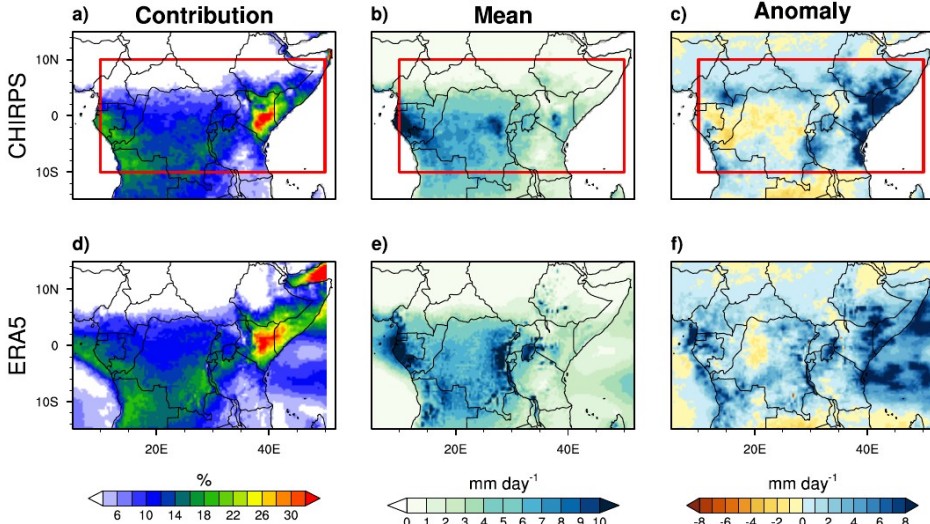

**Fig 1:** (a, d) Percentage contribution of November to total annual rainfall over tropical Africa, (b, e)
LTM (1981-2022) rainfall for November and (c, f) November rainfall anomalies. Rainfall anomalies
are calculated as the difference between CHIRPS 2023 and CHIRPS LTM rainfall.

**4. Identifying important drivers of extreme November 2023 rainfall**
**4.1. Sea Surface Temperatures**

The tropical SST anomalies for November 2023, as estimated by ERSST and ERA5,
are shown in Fig. 2. In both datasets, SST anomalies were predominantly positive
throughout the equatorial and subtropical regions of the Pacific and Atlantic Oceans.
Pacific anomalies were strong and positive over the equator, where anomalies were on
the order of 2 to 2.9 K. That, and the positive and significant correlations with SSTs in
Niño-3.4 (black box in Fig. 3c) indicate that ENSO was a factor in the East African rainfall
anomalies of November 2023 (Chobo and Huo 2024). This ENSO observation occurs when
the Indian Ocean features a strong dipole pattern, with positive anomalies in the west
pole (10° S-10° N and 50°-70° E) and negative in the east pole (10° S-0° and 90°-110° E).
Many studies show that positive ENSO phases are usually associated with positive rainfall
anomalies over East Africa (Indeje et al. 2000; Shilenje and Ogwang 2015; Onyutha 2016).
However, during the two positive El Niño events, 1983 and 1992, East African countries
experienced significant droughts. These previous studies also found that in addition to
ENSO, IOD plays an important role in the modulation of precipitation in East Africa. During
these two years, IOD was in its negative phase. This shows that it is important to take into
account the combined influence of ENSO and IOD in the modulation of precipitation over
East Africa before and during the OND season. In this line, a recent study by Roy and
Troccoli (2024) showed that when IOD and ENSO are both positive in July-August-





September (JAS) and OND, this leads to an increased rainfall over East Africa in OND. Note
that in JAS 2023, these two modes were in their positive phase
(https://origin.cpc.ncep.noaa.gov/products/analysis_monitoring/ensostuff/ONI_v5.php;
https://ds.data.jma.go.jp/tcc/tcc/products/elnino/iodevents.html). An important remark is
the similar patterns in Niño-3.4, IOD, NTA and ATL areas with those in October-November
2019, which were also associated with increased rainfall over East Africa (Nicholson et al.
2022; Ingeri et al. 2024).

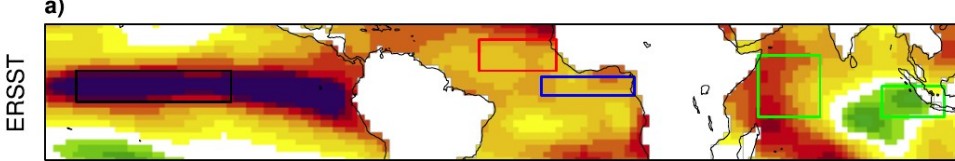

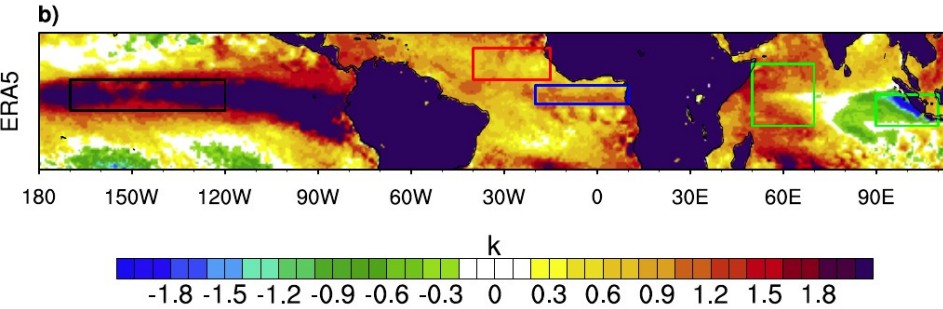


**Fig 2:** SST anomalies during November 2023 for (a) ERSST and (b) ERA5. Red and blue boxes
indicate the areas for north tropical Atlantic (NTA; 5°-15° N and 40°-15° W) and equatorial Atlantic
(ATL; 3° S-3° N and 20° W-10° E) SST calculations, green boxes indicate the areas for DMI
calculation. The black box over the Pacific Ocean is Niño-3.4 (5° S-5° N and 170°-120° W).

The time series of SSTs for the Niño-3.4 region over the period 1981-2023 (black
line in Fig. 4c) indicates the warmth in November 2023. During this period, the 2023
anomalies ranked among the three warmest years on record. This Pacific sector exhibits
the strongest correlation with November rainfall over the CB (Moihamette et al. 2022) and
over eastern EA (Fig. 3c; Palmer et al. 2023; Chobo and Huo 2024; Roy and Troccoli 2024).
These SST anomalies likely played a key role in the positive rainfall anomalies observed in
eastern EA, especially in coastal areas, as well as in the CB. Roy and Troccoli (2024) showed
that when IOD and ENSO are both positive in November, this contributes to excessive
rainfall over the whole of East Africa, south of the DRC, north of Angola, Nigeria and CAR,
while deficit rainfall occurs over certain parts of DRC. In this line, the study by Moihamette
et al. (2022) showed that during simultaneous both positive IOD and ENSO events, the
influence of the positive phase of IOD on EA rainfall is significant with the non-El Niño




effect and this is characterized by an increase in moisture advection toward EA that
contributes to an enhancement of rainfall intensity, more pronounced over eastern and
western EA.

Extreme positive values were recorded in November 1997, 2006, and 2019, all of
which were exceptionally wet years in eastern EA. The 2023 positive dipole event ranks as
the third strongest for November since 1981. Notably, the DMI magnitude in 2023 was
smaller than in both November 1997 and 2019, when conditions in EA were considerably
drier than in 2023, suggesting that additional factors may have contributed. The event
may have influenced the CB, as significant correlations between the DMI and rainfall are
evident (Fig. 3b), with the strongest impact observed in the far eastern EA region (Dezfuli
and Nicholson, 2013). However, rainfall anomalies within CB exhibit both positive and
negative values, which can be linked to the IOD due to significant correlations over the
IOD regions. While heavy rainfall in both CB and eastern Africa is likely associated with the
IOD, this does not hold true for other affected areas. Following Dezfuli and Nicholson
(2013), correlations between SST and rainfall suggest that November rainfall in western EA
is not influenced by the IOD, a conclusion further reinforced by the correlations in Fig. 3a.

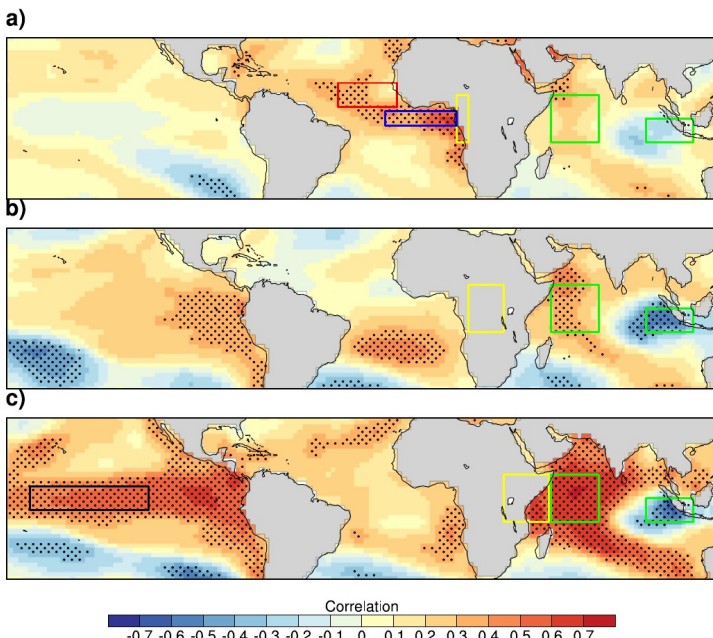


**Fig 3:** Correlation coefficient between (a) Western EA (10° E-15° E), (b) Congo Basin (CB;15° E-30° E),
and (c) Eastern EA (30° E-50° E) rainfall with SST during 1981-2023 period. The stippling occurs
where the correlation is statistically significant at the 95% confidence level through the Student's t
test





The intense rainfall in western EA can likely be attributed to the Atlantic SST
anomalies. It is important to highlight that SSTs along the eastern Atlantic coast and in the
central equatorial Atlantic show a strong positive correlation with November rainfall in
western EA, as reported by Lutz et al. (2014). Figure 3a shows correlations of western EA
November rainfall with SSTs. Over the Atlantic Ocean, significant and positive correlations
between western EA rainfall and SST occur over NTA, ATL and eastern Atlantic coastal
regions (Fig. 3a). Moihamette et al. 2024 suggest that in November, the Atlantic Ocean has
a significant influence during positive IOD events, induced by its teleconnection with the
Indian Ocean. This is characterized by anomalous westerly winds over the central
equatorial Atlantic Ocean (ATL, blue box in Fig. 3a), generating moisture advection
towards western EA. These winds originate from the NTA domain (red box in Fig. 3a).
Furthermore, these ocean regions feature strong standardised SST anomalies in 2023 (Fig.
4b). A feature to note is the exceptional warmth of SSTs in November 2023 over the NTA
area (red line in Fig. 4b). Over the 1981-2023 period, the 2023 anomalies were the
warmest on record. Another feature of NTA variability is its positive correlation (r > 0.2)
with eastern EA rainfall (Fig.3c). A recent study by Ingeri et al. (2024) showed that positive
SST anomalies in NTA from October to December lead to enhanced East African rainfall,
mainly over Tanzania, Kenya and southern Uganda. In addition, the SST time series over
the equatorial Atlantic showed that the 2023 anomalies were the second warmest year
after the November 2019 anomalies which led to a significant increase in rainfall over
western EA (Nicholson et al. 2022; Kenfack et al. 2024).

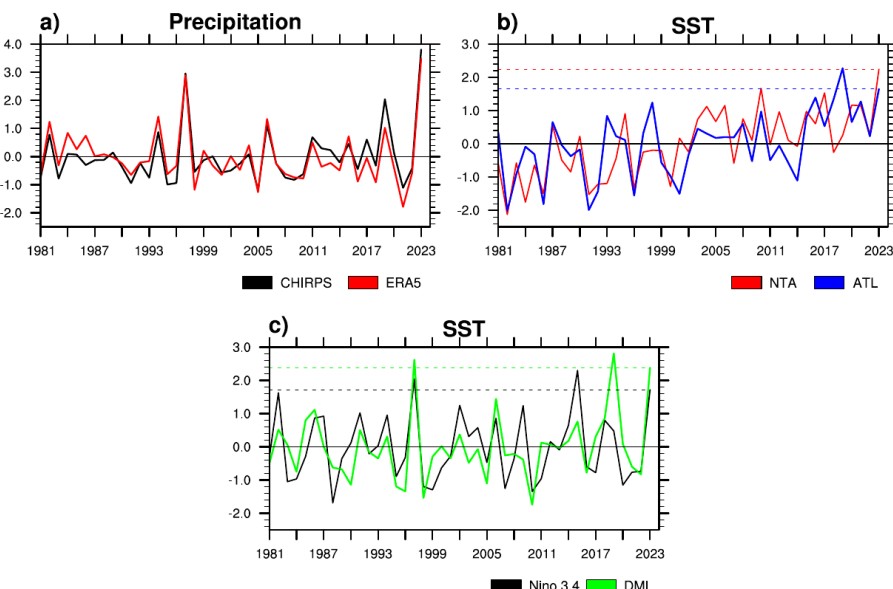


**Fig 4:** (a) Indices of standardised rainfall anomalies over 1981-2023, averaged over the red box
indicated in **Fig. 1**. (b) As in (a), but averaged for the SST over the NTA and ATL oceanic regions. (c)
As in (a), but averaged for the DMI and Niño-3.4.




**4.2. Dynamic factors associated with the November 2023 extreme rainfall**

Many studies (eg., Pokam et al. 2012, 2014; Taguela et al., 2022) proposed a physical mechanism for modulating the long rains over CB on interannual timescales. They identified that changes in the low-level westerlies (LLW) over the equatorial Atlantic Ocean play a significant role in this modulation. These westerlies influence the moisture transport and atmospheric conditions over the region (Nana et al. 2023), impacting the onset, intensity, and duration of the long rains. Variations in LLW can alter the regional climate patterns, and this mechanism helps explain the variability of the long rains in relation to other climatic phenomena, such as the ENSO.

The maximum moisture convergence over equatorial central Africa in SON is found to be a consequence of low-level moisture advection from the Atlantic Ocean. These LLWs are controlled by the heating contrast between cooling associated with subsidence over the ocean and heating over land regions, where ascent prevails. These heating contrasts lead to a Walker-type circulation over the Atlantic Ocean and equatorial central Africa with the development of LLW as its lower branch. These LLW over the equatorial Atlantic are strongly correlated with DMI (Moihamette et al. 2024). For East Africa rainfall variability, numerous studies have identified ENSO and IOD as the two main atmospheric drivers that influenced the October-November-December (OND) rainfall (Indeje et al. 2000; Shilenje and Ogwang 2015; Roy and Troccoli 2024). Following Black (2005), these drivers play an important role in the moisture convergence over East Africa through moisture advection from the Indian Ocean, even if they are not independent of each other. Roy and Troccoli (2024) showed that when ENSO and IOD are in the same phase during JAS, The Walker-like circulation appears to play a major role in modifying the ascending branch into a descending branch in two situations (positive and negative phases) during OND period.

**4.2.1. Zonal circulation/winds, Walker circulation and water vapor mass transported over tropical Oceans**

Following the recent study of Longandjo and Rouault (2024), rainfall variability over EA is highly dependent on moisture inputs linked to atmospheric circulation. It is therefore important to show the characteristics of these moisture inputs to identify their oceanic origin.

Figure 5 depicts the vertical profile of specific humidity (first row) and zonal moisture flux (shaded) overlaid by zonal wind (contour; second row) in November for the climatology of 1981-2022 (Fig. 5a,d), 2023 average (Fig. 5b,e) and the anomaly (Fig. 5c,f), averaged between 10° S-10° N.

Over the Indian Ocean, the 1981-2022 climatology is characterized by intense westerly flux, whereas the November 2023 average appears to be an easterly flux (Fig. 5d,e). Over the continent, moisture flux is predominantly easterly or westerly during November 2023 as in the climatology. We can conclude from this that two distinct



mechanisms govern the moisture flux over the Indian Ocean and the continent. It is
noteworthy that the anomaly analysis (Fig. 5c) confirms that there was more moisture
over the EA in November 2023 than in the 1981-2022 climatology. This moisture surplus
appears here to extend up to 400 hPa and is more pronounced over eastern EA between
900 and 500 hPa.

In addition, from the Atlantic Ocean, the westerly wind in 2023 was more
pronounced near 30° E compared to the climatology (near 17° E). These results lead to a
strong moisture flux from the Atlantic (Indian) Ocean over western (eastern) EA (Fig. 5f),
following the work of Chadwick et al. (2016) who showed that increased humidity over
land would be a response to increased moisture advection from the oceans under
warming. The easterly flux over eastern EA is strong at the middle troposphere, where
strong moisture is observed. It is interesting to note that the intensity of the LLW
anomalies appears to extend to the upper troposphere. In conclusion, the intensification
of the zonal wind over EA indicates that this moisture (strong from 900 hPa) probably
comes from both the equatorial Atlantic and Indian Oceans.

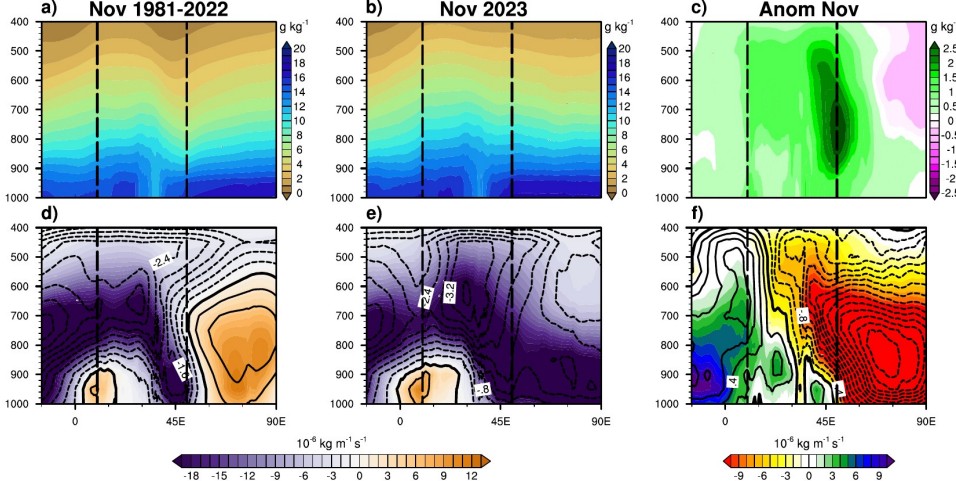


**Fig 5:** Longitude-height cross-sections of (first row) specific humidity and (second row) zonal
moisture flux (shaded: $10^{-6}$ kg m$^{-1}$ s$^{-1}$) and zonal wind (contour: m s$^{-1}$) for (a,d) the climatology of
November 1981-2022, (b,e) November 2023 and (c,f) the anomaly, averaged between 10° S-10° N.

The precedent anomaly analysis leads us to conclude that both the Atlantic and
Indian oceans contribute to the moisture increase (so more rainfall in November 2023)
over the EA and that ENSO has a possible contribution mainly over the eastern EA. This
suggests large-scale tropical climate control. To examine the links with tropical circulation,
large-scale wind, omega and water vapor mass transported field analyses are carried out,
and the results are present in Figures 6-9.



From the lower to the high troposphere, anomalous westerly (easterly) winds
across the equatorial Atlantic (western Indian) region penetrate the EA domain through its
western (eastern) border (Fig. 6a-c). These westerly winds come from the NTA region (at
the low troposphere, Fig. 6a).

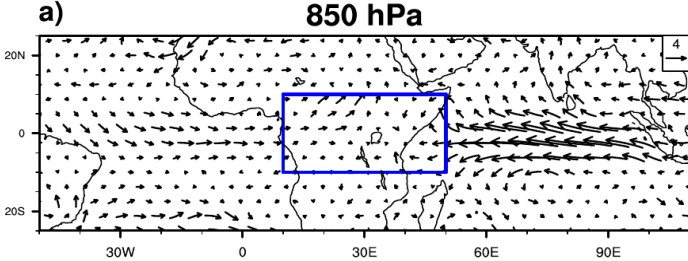

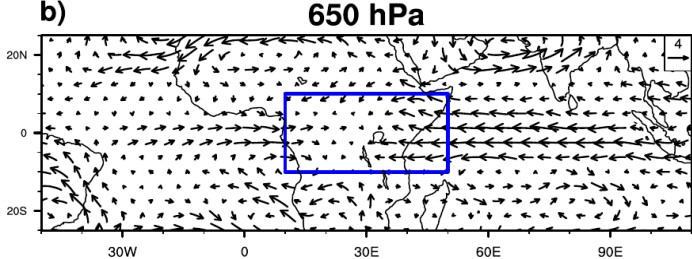

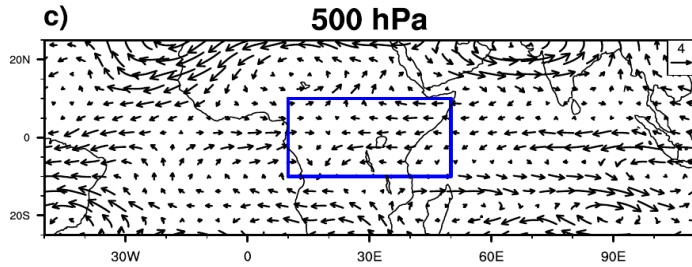

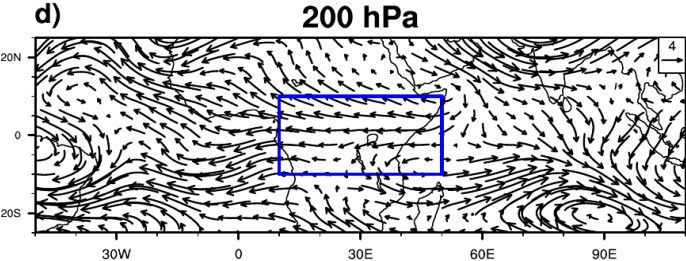

**Fig 6:** Vector wind anomalies (Nov. 2023 minus 1981-2022 mean) at (a) 850 hPa, (b) 650 hPa, (c) 500
hPa and (d) 200 hPa.

13                            13





398   In addition, over both the equatorial Pacific and Atlantic Oceans, and western EA,
399 westerly anomalies feature at 850 hPa while easterly anomalies are evident over the
400 equatorial Indian Ocean (Fig. 7c). Inverse anomalies are observed at 200 hPa (Fig. 7b),
401 suggesting that the west-east zonal circulation is subject to changes. These later are
402 shown by the upper-level divergence and vertical motion analysis. Equatorial Africa and
403 the coasts of the Atlantic and Indian Oceans feature strong divergence at 200 hPa,
404 followed by strong convergence over the NTA and eastern Indian Ocean regions (Fig. 7a),
405 the reverse divergence anomalies' pattern characterises lower-troposphere level (not
406 shown). These patterns are consistent with Walker circulation, which has been examined
407 by both vertical velocity (omega), and zonal wind combined with vertical velocity (Fig. 8).
408 Note that negative values of omega denote ascent motions and positive values indicate
409 subsidence.

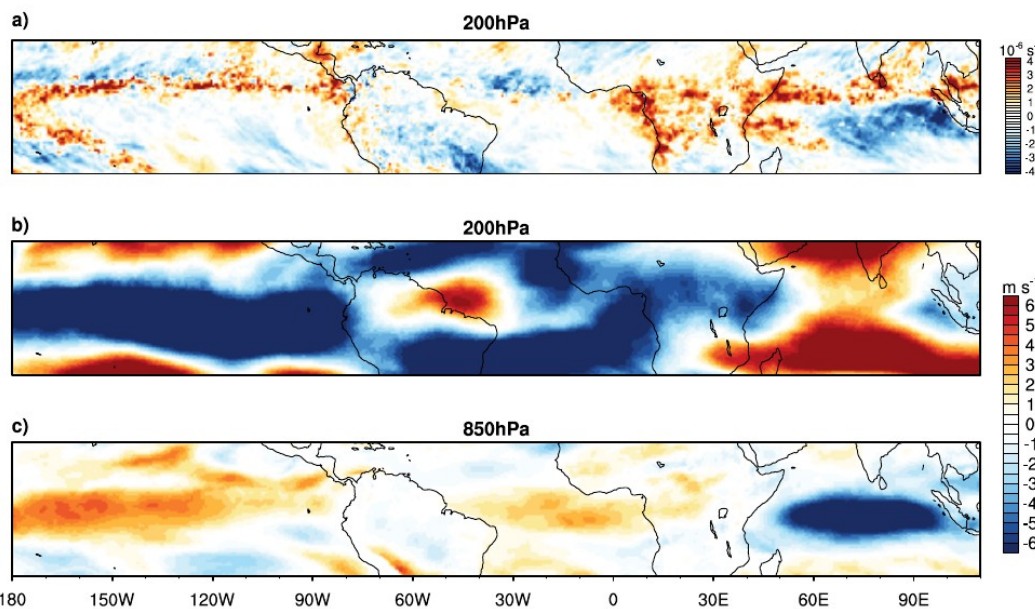

412 **Fig 7:** Anomalies of (a) divergence ($10^{-6}$ s$^{-1}$) at 200 hPa and zonal wind (m s$^{-1}$ ) at (b) 200 hPa and (c)
413 850 hPa.

415   For the 1981-2022 climatology (Fig. 8a), Omega indicates that the ascent motions
416 of the Indian Ocean Walker cell are very pronounced. The western Atlantic Ocean is
417 characterized by strong rising motions, while the eastern Atlantic Ocean experiences
418 sinking motions. Over western Africa (10° E to 30° E), the entire atmospheric column
419 shows significant upward motion, whereas in eastern Africa (30° E to 45° E), rising
420 motions dominates at low levels, while sinking motions prevails in the mid- and upper
421 troposphere. This subsidence in eastern Africa leads to reduced rainfall. These findings
422 are consistent with those observed during the exceptional October/November 2019



events, as noted by Nicholson et al. (2022). By contrast, these branches are weak in
general during November 2023 (Fig. 8b). In contrast to climatology, 2023 shows strong
ascendance at mid- and upper-troposphere over eastern Africa.  Here, we focus on three
omega (color), and zonal wind combined with vertical velocity (vector) patterns to evaluate
the location and strength of the African Walker circulation cells. Anomalous rising motion
corresponds to areas with low-level converging vectors, mid-level ascent motions and
upper-level diverging vectors that will typically experience more rainfall. Following Fig. 8c,
at 850 hPa, anomalous rising (sinking) motions are associated with areas of westerly
(easterly) anomalies. At 200 hPa, anomalous ascendance (subsidence) corresponds to
areas of easterly (westerly) anomalies.
It is noteworthy that rainfall deficits are observed over the Congo Basin, around 15-
35° E. This region of rainfall deficits is linked to a corresponding area of reduced rising
motion at low levels (Fig. 8c). Over the three oceans, the zonal cells are weaker, but more
pronounced over the Atlantic and Indian oceans than the Pacific ocean. In the case of the
Pacific Ocean, the increase of Pacific cells is linked with the El Niño events and are moved
westward and vertical motion anomalies are weak along the coast of South America. Thus,
the SST's El Niño pattern could be highly developed. Both the Atlantic and Indian Oceans
feature greater vertical motion contrasts than the Pacific Ocean. This is characterised by
an increased ascent (subsidence) over the eastern and western Atlantic and Indian oceans
respectively, and increased subsidence (ascent) over the western and eastern Atlantic and
Indian oceans respectively. The consequence is a strengthening of the ascent which
extends over equatorial Africa and is accompanied by an increase in rainfall over Africa,
mainly in East Africa.



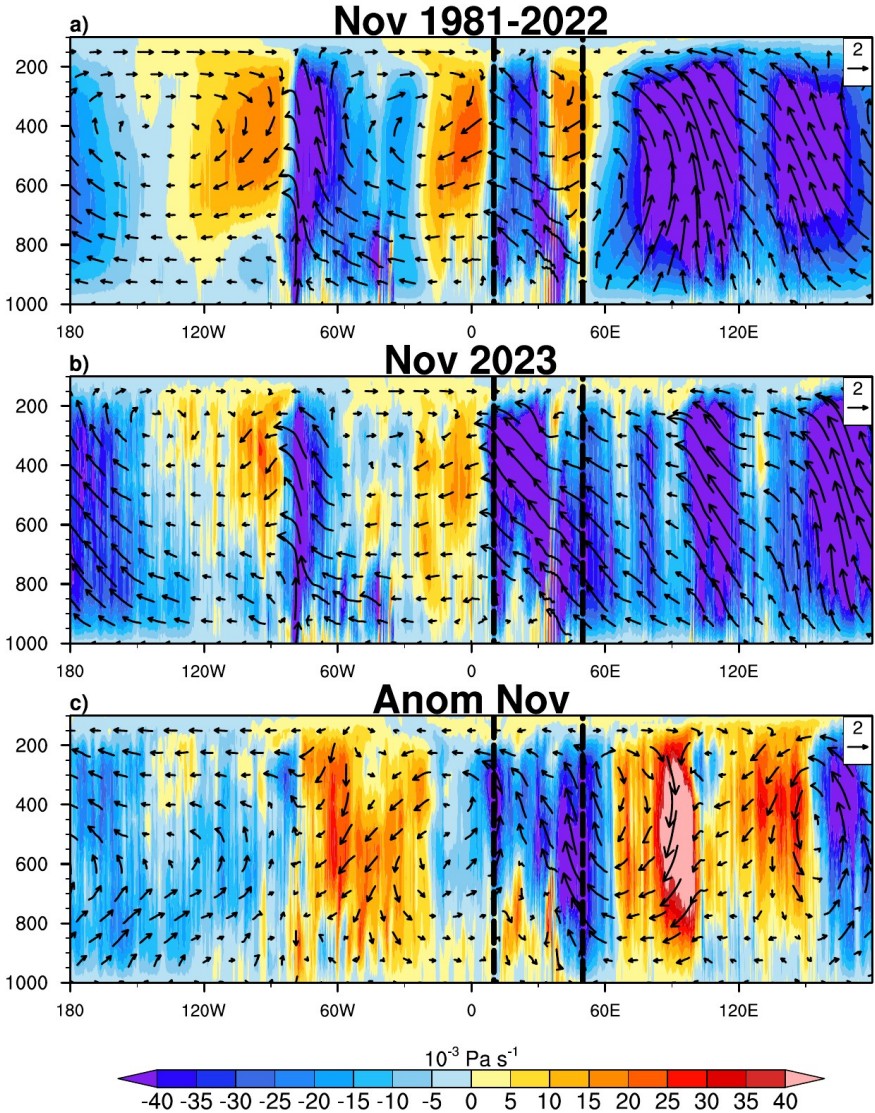


**Fig 8:** Longitude-height cross-sections of vertical velocity (Omega: $10^{-3}$ m s$^{-1}$) selected over latitude
10° S-10° N in November for (top to bottom) climatology, November 2023, and anomaly. The
shading (background) represents the vertical velocity, with negative values denoting ascent
motions and positive values indicating subsidence. Additionally, the vectors (overlay), obtained with
the zonal wind component and vertical velocity, illustrate the behavior of the air mass during
upward or downward movements


To further investigate the vertical motion, the water vapor mass transported
analysis is done through the mass-weighted stream-function (Fig. 9). In the climatology
mean (Fig. 9a), the CB cell ($\psi < 0$) is located between 1° and 18° E and extends up to 950



hPa, whereas it occurs at -1 and 28° E and extends around 975 hPa during November
2023 (Fig. 9b). These CB cell locations coincide with the Walker cell ascending branch (Fig.
8a,b) over EA, more intense during November 2023 associated with sinking branches over
the Equatorial Atlantic and Indian Oceans (Figs. 8c and 9c). Near to 800 hPa, the westward
($\psi > 0$) of the circulation is greater during November 2023 (Fig. 9c), leading to the
presence of easterly Jet at the middle-tropospheric. These results confirm those obtained
by the Walker circulation and, consequently, the pattern of rainfall anomalies.



**Fig 9:** Zonal mass-weighted stream function (contours: $10^{11}$ kg s$^{-1}$) averaged between 10° S-10° N
for (a) mean, (b) Nov. 2023 and (c) Nov. 2023 minus mean. Positive (negative) values indicate the
westward (eastward) circulation.
**4.2.2. Moisture flux divergence**
Figure 10 shows the vertically integrated (1000-200 hPa) moisture flux convergence
(color) and vertically integrated moisture flux (vectors). Positive values indicated
convergence and negative values indicated divergence. Overall, the whole of EA exhibits
moisture convergence with strong and significant moisture convergence in areas which
feature strong and positive rainfall anomalies (Fig. 1c,f). The weak moisture divergence
observed over the central part of DRC confirms the weak and negative rainfall anomalies
also observed in this area. Note that all vectors in the EA region originate from the two
neighbouring oceans through the western and eastern boundaries only. No vectors have
entered the region across the northern and southern boundaries. This confirms that these
two oceans were mainly responsible for the wet episodes of 2023. The strong westerly
winds become southwesterly to easterly, advecting moist air from the Atlantic Ocean
towards the northern regions (Gabon, northern DRC, CAR and Cameroon). It is
noteworthy that the strong westerly winds over the equatorial Atlantic originate from the
NTA region, where strong and significant moisture divergence has been observed (not
shown). For the eastern EA boundary, strong easterly winds from the Indian Ocean advect
moist air in eastern and some southern regions, mainly over Somalia, southwest Ethiopia,
eastern Kenya, Uganda and northern Angola. These results confirm the warm SST feature
over oceans in Figure 2. Although the November 2023 DMI is lower than in 2019 when the
strongest positive IOD event since the 1950s occurred during October and November



(Nicholson et al. 2022), the eastern region of EA was wetter in 2023 compared to 2019.
This can be seen through the convergence and moisture flux anomalies, and
consequently the higher precipitation in November 2023. One explanation could be the
significant presence of the El Niño event in 2023, which contributed to humidifying the
Eastern Africa region (Palmer et al. 2023; Roy and Troccoli 2024), unlike 2019, when during
the positive IOD event, the El Niño episode was absent (Nicholson et al. 2022).

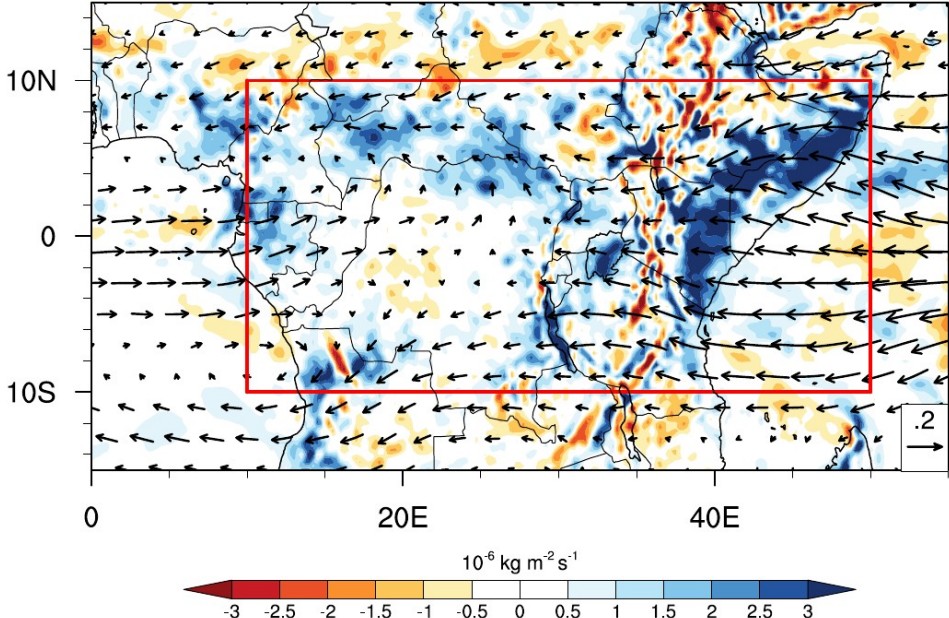

**Fig 10:** (a) Anomalies of vertically integrated (1000-200 hPa) moisture flux (vectors: kg m$^{-1}$ s$^{-1}$) and
vertically integrated moisture flux convergence (positive values) or divergence (negative values)
anomalies (shading: 10$^{-6}$ kg m$^{-2}$ s$^{-1}$). Only significant vectors (shading) above the 90 % (95 %) level
are shown

The northern regions (western Nigeria, northern Cameroon, southern Chad, CAR
and South Sudan), which receive less than 2 mm day$^{-1}$ of rainfall and have a near zero
percentage contribution of November rainfall (Fig. 1a-b, d-e), recorded heavy rainfall in
November 2023 (Fig. 1c-f). Similar to Nicholson et al. (2022) in October 2019, Figure 11a
confirms this northward displacement of the state of the African monsoon in November
2023. During this year, the 35 mm isopleth of the total column water vapor and the
intertropical discontinuity (dashed blue and red lines respectively) move to the north,
enhancing rainfall in the northern regions. These dashed lines correspond to the limit of
the anomalous meridional mean sea level pressure gradient, characterised by anomalous
high pressure over the south of the 35 mm isopleth of the total column water vapor and
the intertropical discontinuity lines, and lower over the north. Except over the Eastern EA




areas. Similar pressure, intertropical discontinuity and 35 mm isopleth of the total column
water vapor, and easterly (westerly) moisture flux anomaly were observed over Eastern EA
(Gabon, CAR and Chad) in October (November) 2019 when extreme rainfall and flooding
occurred in the latter country (Nicholson et al. 2022). As shown in Figure 11b, low-cloud
cover (LCC) is below average in the southwest of  EA (mainly over CB and south of
Cameroon and Gabon) but becomes above average over the northern part of EA (mainly
over CAR) and the whole of East Africa. These positive LCC anomalies coincide with
positive and strong surface net solar radiation anomalies. These spatial differences in
atmospheric convective activity revealed by LCC and surface net solar radiation explain
the spatial variations in rainfall anomalies shown in Fig. 1c,f.

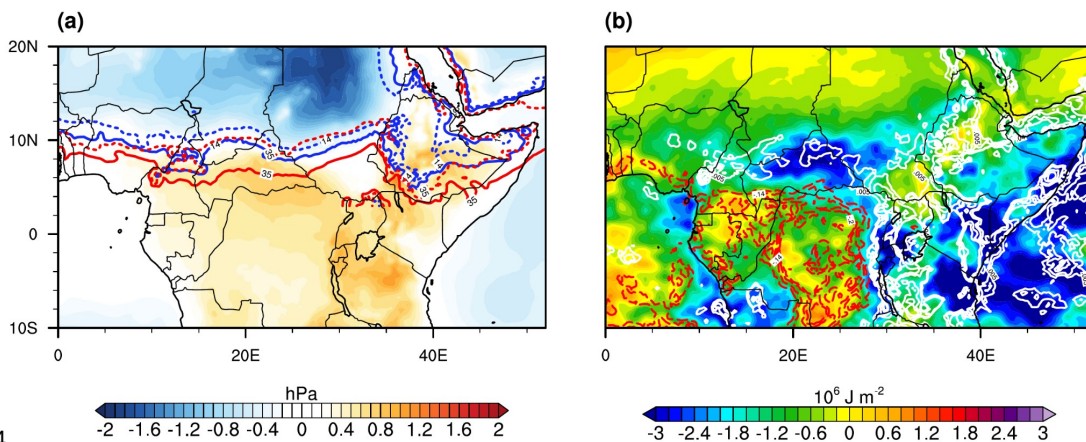

**Fig 11:** (a) The anomalous state of the African monsoon in November 2023 is characterized by
mean sea-level pressure anomalies (colored) relative to the 1981-2022 average, along with the
positions of the intertropical discontinuity (ITD) and total column water vapor isopleths. The ITD is
indicated by the 14°C 2-meter dew point temperature contours, with the mean ITD position shown
by a solid blue line and the November 2023 position by a dashed blue line. Similarly, the mean
location of the 35 mm total column water vapor isopleths is marked by a solid red line, while the
November 2023 position is represented by a dashed red line. (b) Spatial representation of the
surface net solar radiation anomalies (shading: $10^6$ J m$^{-2}$). Red (white) lines represent negative
(positive) low cloud cover (%) anomalies.

### 4.2.3. LLWs, easterly Jets and MJO activity

One of the key atmospheric features over western EA are the African Easterly Jet
(AEJ) and Tropical Easterly Jet (TEJ). Following Nicholson and Grist (2003), AEJ and TEJ are
maximum easterly winds that occur at the mid-troposphere (from 700 to 600 hPa) and
upper-troposphere (around 200 hPa) respectively. Here, we describe the characteristics of
these atmospheric features during the Nov 2023 extreme rainfall. The African Easterly Jet's
southern component (AEJ-S) only appears from September to November and its jet core is



located around 10° S in November (Kuete et al. 2022). In contrast to the AEJ-S, the northern component (AEJ-N) occurs during all months of the year, and its core is located around 5° N in November. As the AEJ-N, TEJ features during all months and is located near the south equator. Nicholson and Grist (2003) showed that Central Africa is a region where the divergence of the upper troposphere (Fig. 7a) is enhanced, which could favour convective activity (Fig. 8c), suggesting that the variability of the TEJ may influence the variability of precipitation in the region. Figure 12 shows Latitude/height cross-sections of easterly winds (dashed contours) of the 1981-2022 November mean and November 2023 at 10° E, overlaid by the zonal moisture flux (color) calculated from the West boundary (10° E) minus East boundary (30°E).

For the mean climatology (Fig. 12a), only AEJ-N is observed around 3°N at 700 hPa, with core speeds reaching 10 m s$^{-1}$. This intensification of AEJ-N coincides with strong moisture flux divergence. During November 2023 (Fig. 12b), both AEJ components and TEJ are present at the mid- and upper-troposphere respectively. During this year, the mid-level zonal moisture convergence (divergence) induced by the weak AEJ-S (jet core not exceeding 7 m s$^{-1}$; Kuete et al., 2019) increases (decreases), favorasing increases mid-tropospheric moisture convergence south of the equator over western EA, resulting in wet conditions over the domain (Fig. 1c,f). AEJ-N moves further north and its intensity decreases from 10 m s$^{-1}$ to 8 m s$^{-1}$, leading also to more western EA rainfall during positive IOD events, more pronounced during the October-November months (Moihamette et al. 2024). Following Moihamette et al. (2024), this decrease of the AEJ-N drove the equatorial easterly moisture transport to the western EA. These conclusions are in agreement with a study by Dezfuli and Nicholson (2013) showing that during October and November, the months with stronger (weaker) AEJ components experience dry (wet) conditions over western EA. Also, Nicholson and Grist (2003) showed that when both AEJ components are present, the western EA's rain-band moves further south (10° S to the equator), this is only observed during November. Another feature is the LLWs, which are weak in climatology, were anomalously strong and extended up to 700 hPa, mainly over southern-hemisphere latitudes. These wind changes lead to enhanced rainfall in western EA during the SON seasons (Pokam et al. 2014; Kuete et al. 2019; Taguela et al. 2022).



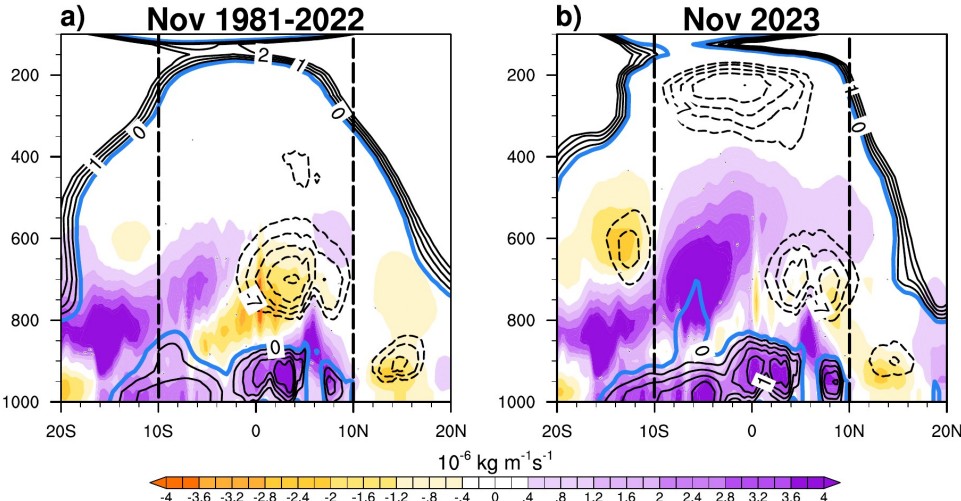

**Fig 12:** Latitude/height cross-sections of net zonal moisture flux ($10^{-6}$ kg m$^{-1}$ s$^{-1}$) calculated from West boundary (10° E) minus East boundary (30° E) for (a) climatology and (b) November 2023. Black solid (dashed) lines represent LLW (AEJ components and TEJ; U<-6 m s$^{-1}$) with the contour interval of 0.5 (1) m s$^{-1}$, calculated at 10° E for the respective periods. Positive values indicate moisture flux convergence, and negative values moisture flux divergence.

Here, we analyse one of eastern Africa's intra-seasonal climate drivers, the MJO, through the Real-time Multivariate MJO (RMM) index. This later strongly influences the equatorial Africa region's rainfall during March-April-May (Sandjon et al. 2012) and November and December (Pohl and Camberlin 2006; Berhane and Zaitchik 2014; Berhane 2016; Zaitchik 2017; Palmer et al. 2023). According to Pohl and Camberlin (2006) and Kimani et al. (2020), during phases 6-8 (1-4) of the MJO, there is an increased chance of the highest rainfall over the coastal (highlands) regions of East Africa during OND months, through moisture advection from the Indian Ocean. The ENSO-MJO relationship was studied by Pohl and Matthews (2007), who found that maximum MJO activity is often observed during El-Niño events. In addition, Berhane (2016) showed that this juxtaposition of MJO activity and ENSO events leads to an increase in precipitation that is greater than when El-Nino events are absent.





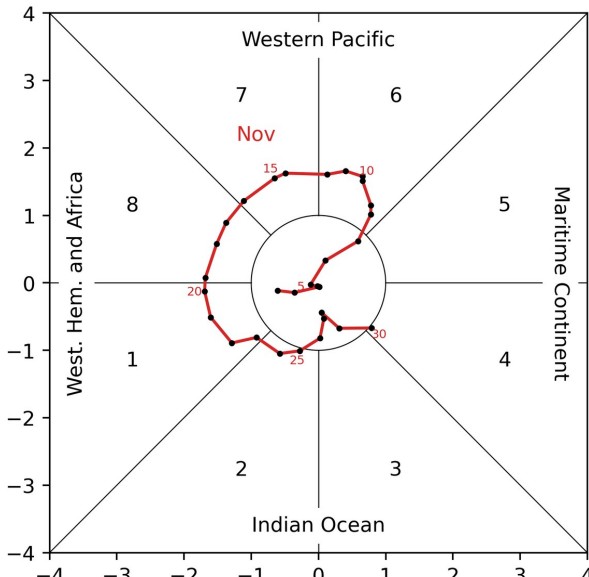

**Fig 13:** Madden-Julian Oscillation (MJO) phases and intensity (red line) space diagram for November 2023. Each black dot represents the value for a given day with select dates labelled in red. [Source: NOAA/NCEI Climate.gov, https://www.ncei.noaa.gov/access/monitoring/monthly-report/synoptic/202311, accessed 26/11/24]

Figure 13 shows the MJO phase space diagram for November 2023, based on the RMM index. It should be noted that the MJO was active for 19 days, during phases leading to wet periods in the highlands and coastal areas of East Africa (Pohl and Camberlin 2006; Berhane and Zaitchik 2014; Zaitchik 2017). During November 2023, the MJO was in phases 6-8 from 8 to 19 November, which are the phases leading to wet spells over the East African coast (Pohl and Camberlin 2006; Zaitchik 2017), mainly over Somalia and the eastern parts of Ethiopia, Tanzania and Kenya regions. Also, during 20-25 and 30 November 2023, the MJO was in phases 1, 2 and 4, which are the phases typically associated with increased rainfall over the Highland region of East Africa (Pohl and Camberlin 2006), mainly over Uganda, the western and northern part of Kenya and Tanzania, respectively.

**5. Summary and Conclusions**

This study examines the extreme wet conditions that occurred in November 2023 in Equatorial Africa (EA) and shows that this rainy episode was caused by several factors. While some anomalous rainfall over both western and eastern equatorial Africa were attributed to moisture transport from the Atlantic and Indian oceans respectively, the unusually very strong November 2023 MJO activity was a significant factor. In addition to meteorological conditions, further research is therefore needed to quantify the roles that



dynamic and thermodynamic processes played in the extreme events of November 2023.
The most important findings of this study are as follows:

- Although the rain band in November is over the south equator, many north areas feature positive rainfall anomalies, most pronounced over eastern Africa. These strongest November 2023 rainfall anomalies occur when significant SST anomalies were observed in the three equatorial oceans.

- In contrast to the extreme rainfall of November 2019 in East Africa, where the DMI reached record levels, the DMI of November 2023, which is lower than that of 2019, is causing more rainfall. This may be due to the presence of strong El Nino conditions over the equatorial Pacific in 2023. But over central Africa, the rainfall anomalies in 2023 are lower than in 2019, certainly due to the state of the equatorial Atlantic Ocean, which was warmer in 2019 than in 2023.

- SST anomalies over the Atlantic (Indian) Ocean are associated with anomalous westerly (easterly) winds that bring more moisture over EA (Fig. 5c). Moisture flux from both oceans, respectively (Fig. 5f), induced a weakening of the ascending and descending branches of both neighboring oceans Walker-like cells (Figs. 8 and 9).

- The westerly moisture flux from the Atlantic Ocean that veered into southwesterlies moved the rainbelt further north by enhancing the transport of moist air over the northern (5°-10° N) regions (Fig. 11).

- Over western EA where extreme rain also occurs, the African wind and easterly wind regime is an important factor. The presence of strong equatorial westerlies, AEJ-S and TEJ, and the movement of AEJ-N further north have also retarded the retreat of the West African monsoon, causing positive rainfall anomalies over northern areas (especially over Nigeria, Cameroon, CAR, Sudan and South Chad).

- Another driver is the MJO activity, which was active during several days of November 2023 (Fig. 12). We have shown that the positive rainfall anomalies over East Africa coincided with active phases of MJO, which enhanced rainfall in November over both western and eastern areas of East Africa. This increase in rainfall was significant with the occurrence of the El-Niño events.

This study demonstrates that the anomalous wetness conditions over Equatorial Africa were caused mainly by the Atlantic and Indian oceans, through the anomalous moisture transport and moisture flux divergence, Walker circulation, and changes in the zonal winds, induced by extremely strong SST anomalies. These anomalous patterns were similar to those observed over this region in October/November 2019 when extreme rainfall occurred, following Nicholson et al. (2021). The present study demonstrates the importance of accounting for ocean-atmosphere interactions in intra-seasonal forecast



models to refine regional climate information provided to policymakers. It is important to
highlight that the robustness of these findings requires additional evaluation. Our study
exclusively examines the meteorological causes of these extreme events. Further
investigations should encompass the roles that dynamic and thermodynamic processes
played in these November 2023 events.



**Code availability** Figures shown in this study are plotted using the NCAR Command
Language (NCL; https://doi.org/10.5065/D6WD3XH5, NCAR Command Language, 2017).
Codes can be obtained from the corresponding author.

**Data Availability Statement** All observational and reanalysis data used in this study are
publicly available at no charge and with unrestricted access. One plot is generated using
the web site of NOAA/NCEI, at https://www.ncei.noaa.gov/. The ERA5 reanalysis is
produced within the Copernicus Climate Change Service (C3S) by the ECMWF and is
accessible via the link https://cds.climate.copernicus.eu/datasets/reanalysis-era5-
pressure-levels-monthly-means?tab=download; the CHIRPS2 data are available at
https://data.chc.ucsb.edu/products/CHIRPS-2.0/global_daily/netcdf/; the ERSST data are
available at https://iridl.ldeo.columbia.edu/SOURCES/.NOAA/.NCDC/.ERSST/.version5/.

**Author's contributions**
**HNN:** Conceptualization; data upload; data analysis; formal analysis; investigation;
methodology; software; validation; writing-original draft; writing-review and editing. **MG:**
Project administration; supervision; formal analysis; investigation; validation; writing-
original draft; writing-review and editing. **RST:** Project administration; supervision;
validation; methodology; writing; review and editing. **ATT:** Project administration;
validation; methodology; writing; review and editing. **DAV:** Project administration;
supervision; validation, methodology; writing; review and editing.

**Conflict of Interest** The authors declare no conflicts of interest relevant to this study.

**Funding** Not applicable

**Acknowledgements** The authors thank you to all reanalysis, observational and satellite
data providers used in this study. We would like to express our gratitude to the
anonymous reviewers, along with the editor for their constructive suggestions, which
have greatly improved the quality of the paper. We gratefully appreciate the efforts of the
International Joint Laboratory Dynamics of Terrestrial Ecosystems in Central Africa (IJL
DYCOCA/ LMI DYCOFAC) initiative during the realisation of this work.



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
