# Peer review of "Diverse Causes of Extreme Rainfall in November 2023 over Equatorial Africa Hermann N. Nana1\* · Masilin Gudoshava2 · Roméo S. Tanessong3,1 · Alain T. Tamoffo4 · Derbetini A. Vondou1 1Laboratory for Environmental Modelling and Atmosphe"

_EGUsphere, 2025_

## Author Comment (AC1)

Dear Reviewer #1 Thank you for your careful review of our manuscript. Your comments are greatly appreciated and we think this new version of the manuscript responds to your concerns and provides an interesting contribution to the study of November 2023 extreme rainfall. The pdf file containing the response is attached.

**Anonymous Referee #1**

SOME GENERAL COMMENTS

*Rainfall over Africa should not be evaluated using a reanalysis product. There are plenty of good observational and satellite products. It is well known that reanalysis products do not adequately represent rainfall in this region, unless the rainfall presented in the reanalysis is merely from an observational or satellite source. A recent article by Lavers et al. shows that ERA5 cannot get maximum precipitation right. Since the authors do use reanalysis rainfall, they need to find out more about the rainfall product in ERA5 and discuss this in the manuscript. They should also find an article (I think one exists) discussing how well ERA5 performs over Africa.*

Response: **We agree with the reviewer on this point. We recognise that ERA5 underestimates rainfall in Africa, however this reanalysis does a good job of representing the seasonal variability of September-October-November rainfall (Kenfack et al. 2024a), mainly over the whole region. Also, Gleixner et al. (2020) showed that although it underestimates rainfall amounts, the ERA5 reanalysis represents rainfall well during extreme years, over east Africa region.**

**Furthermore, given that this study focuses on the causes of the November 2023 rainfall, it was imperative for us to focus on dynamic analyses, which is why the ERA5 reanalysis outputs were used, following the study by Cook and Vizy (2021) which demonstrated that ERA5 reanalysis is able to represent precipitation regime and associated dynamics compared with other reanalyses, especially over the Congo Basin. We drew up this rainfall anomaly map with ERA5 to check that the reanalysis simulates extreme rainfall, in order to study how the dynamic fields are associated with this rainfall pattern.**

**However, we repeated these analyses with the TAMSAT, GPCP, GPCC and IMERG datasets, and the results are similar. Thus, we have concluded to present only the CHIRPS results. below the results with other datasets.**

[Figure]

Kenfack, K., Tamoffo, A. T., Tchotchou, L. A. D., Marra, F., Kaissassou, S., Nana, H. N., & Vondou, D. A. (2024). Processes behind the decrease in Congo Basin precipitation during the rainy seasons inferred from ERA-5 reanalysis. *International Journal of Climatology*, *44*(5), 1778-1799. https://doi.org/10.1002/joc.8410

Gleixner, S., Demissie, T., & Diro, G. T. (2020). Did ERA5 improve temperature and precipitation reanalysis over east africa? *Atmosphere*, *11*(9), 996. https://doi.org/10.3390/atmos11090996

Cook, K. H., & Vizy, E. K. (2021). Hydrodynamics of regional and seasonal variations in Congo Basin precipitation. *Climate Dynamics*, *59*(5-6), 1775-1797. https://doi.org/10.1007/s00382-021-06066-3

There is no monsoon over East Africa. There are some studies that suggest there is, but only monsoonal wind shift is in the two dry seasons. The only monsoonal area is West Africa. This should be removed from line 82 and also the caption of Fig. 11a, and wherever else it appears.

**Response:** **We agree with the reviewer that the presence of Monsoon in East Africa is the subject of many debates. We have rewritten these sentences.**

The article omits what this reviewer considers to be pivotal articles on variability of the short rains, Hastenrath et al. 2010 and 2011. These clearly explain the importance of the low-level wind anomalies seen in Fig. 6.

**Response:** **This comment is taken into account. Please see the revised manuscript. Line 412**

*The recent paper by Herrnegger et al. (2024) discussing the flooding in 2023 should also be added when discussing the rainfall anomalies.*

**Response:** **This comment is taken into account. Please see the revised manuscript. Line 90**

**A LOT OF SMALL ISSUES**

*Line 115 - Nicholson (2015) demonstrated that the IOD, ENSO, and zonal winds all play a role; did not state that increased rainfall is due to the presence of the IOD. This whole discussion is confusing. All three of those factors play a role. They major occur jointly, but each alone can also produce increased rainfall.*

**Response:** **We agree with the reviewer on this point. In fact, the sentence is not complete. It has been rewritten. Please see the revised manuscript in Lines 116-119**

**"Studies by Nicholson (2015) showed that IOD plays a role in the East African rainfall modulation, while Palmer et al. (2023) showed that increased rainfall in this region is due to the presence of positive IOD events in the October-December (OND) season."**

*Line 170 the word "more" should be replaced by "additional"*

**Response:** **This comment is taken into account. Please see the revised manuscript in Lines 173**

*Line 189 Perhaps I missed it, but I don't think CB cell has been defined.*

**Response:** **This comment is taken into account. Please see the revised manuscript and the text adjusted. Lines 192-205**

"**Research indicates the existence of a shallow, zonal overturning circulation over western EA, identified and termed the Congo Basin cell (CB cell) by Longandjo and Rouault (2020). This cell is a closed, and shallow zonal circulation confined to the lower troposphere (1000-800 hPa), and remains active throughout the year. Similar to Low-level westerlies (LLWs), the CB cell's intensity and width are influenced by near-surface temperature warming over both the western EA landmass and the eastern equatorial Atlantic (Longandjo and Rouault 2020; Taguela et al. 2022). The cell reaches its peak intensity and width in September. According to Longandjo and Rouault (2020), the CB cell's eastern boundary aligns with the Congo Air Boundary, a convergence zone. Here, LLW originating from the equatorial Atlantic, after traversing western EA, converges with the easterly winds of the Indian monsoon system, creating the cell's ascending branch. This convergence zone is characterized by peak convection and precipitation.**

**Consequently, the longitudinal position of maximum rainfall in the region is determined by the width of the CB cell."**

*Fig. 1a and d -this calculation must be off. November cannot possibly supply more than 30% of annual rainfall over Kenya and southern Somalia.*

Response: **This calculation was repeated using other datasets (gauge, satellite and operational) and the results are the same. Furthermore, the method used to carry out this calculation is the same as that used by Gudoshava et al. (2022), where the authors showed that over central Kenya (south of Somalia), rainfall during the OND season has a contribution of more than 60% (50%), and Palmer et al. (2023) showed that peak rainfall during this season occurs during the months of October-November.**

Gudoshava, M., Wanzala, M., Thompson, E., Mwesigwa, J., Endris, H. S., Segele, Z., Hirons, L., Kipkogei, O., Mumbua, C., Njoka, W., Baraibar, M., de Andrade, F., Woolnough, S., Atheru, Z., & Artan, G. (2022). Application of real time S2S forecasts over Eastern Africa in the co-production of climate services. *Climate Services*, *27*, 100319. https://doi.org/10.1016/j.cliser.2022.100319

Palmer, P. I., Wainwright, C. M., Dong, B., Maidment, R. I., Wheeler, K. G., Gedney, N., Hickman, J. E., Madani, N., Folwell, S. S., Abdo, G., Allan, R. P., Black, E. C. L., Feng, L., Gudoshava, M., Haines, K., Huntingford, C., Kilavi, M., Lunt, M. F., Shaaban, A., & Turner, A. G. (2023). Drivers and impacts of Eastern African rainfall variability. *Nature Reviews Earth & Environment*, *4*(4), 254–270. https://doi.org/10.1038/s43017-023-00397-x

[Figure]

*2.1 There is a units problem. SSTs should not be in K. It should be kg, not Kg. Surface pressure should be hPa, not Pa.*

**Response: Here, we have just set the SSTs and Surface pressure units as they are indicated when they are downloaded from the copernicus library ([https://cds.climate.copernicus.eu/datasets/reanalysis-era5-single-levels-monthly-means?tab=overview](https://cds.climate.copernicus.eu/datasets/reanalysis-era5-single-levels-monthly-means?tab=overview)). We have rewritten the 'Kg' unit as 'kg'. Please see line 152 in the revised manuscript.**

*Line 242 This is very misleading. Rainfall was just below normal in 1992. Even 1983 I would not call a drought year.*

**Response: Several authors (e.g., Shisanya 1990) have shown that during these two El Nino years (1983 and 1992), several regions of East Africa (east of 35° E) experienced severe droughts, such as Kenya, which was one of the countries most affected by the 1983 drought (Shisanya 1990), and Tanzania in 1992 (Ibebuchi 2021). Fig. 4a shows the standardised November rainfall anomalies not just over eastern Africa (35°-50° E), but over the whole of Equatorial Africa (10°-50° E), which explains why the negative anomalies are small. The references to these statements have been added to the revised manuscript for the reader's information. Lines 223, 260**

Ibebuchi, C.C., 2021. Revisiting the 1992 severe drought episode in South Africa: the role of El Niño in the anomalies of atmospheric circulation types in Africa south of the equator. Theor. Appl. Climatol. 146, 723-740.

Shisanya, Chris A., 1990. The 1983-1984 drought in Kenya. J. East. Afr. Res. Dev. 20, 127–148. [http://www.jstor.org/stable/24326214](http://www.jstor.org/stable/24326214)

*Fig. 2. The authors need to look further at how ERA5 obtains SST data. Surely ERSST is incorporated into it, which would explain the similarities in a and b. It might just use ERSST. Again, as with rainfall, the authors should have used a bona fide SST data set.*

**Response: Compared to other reanalysis datasets, ERA5 better captures the SST over the tropical Oceans (Yao et al. 2021). The study by Huang et al. (2018) showed that ERSSTv5 and HadISST perform well for SSTs over the Pacific, Atlantic and Southern Oceans, with better performance in the ERSSTv5 dataset. However, we repeated these analyses with the HadISST and OISSTv2 datasets, and the results are similar. Thus, following Huang et al. (2018), we have concluded to present only the ERSST results. below the results with other datasets.**

[Figure]

Huang, B., Angel, W., Boyer, T., Cheng, L., Chepurin, G., Freeman, E., Liu, C., & Zhang, H.-M. (2018). Evaluating SST analyses with independent ocean profile observations. *Journal of Climate*, *31*(13), 5015–5030. https://doi.org/10.1175/jcli-d-17-0824.1

Yao, L., Lu, J., Xia, X., Jing, W., & Liu, Y. (2021). Evaluation of the ERA5 sea surface temperature around the Pacific and the Atlantic. *IEEE Access*, *9*, 12067-12073. https://doi.org/10.1109/access.2021.3051642

*Fig. 3 needs more in the caption. For example, what are the boxes? Indicate that the correlation is with November rainfall. This is clarified later on, but this should be put in the caption at this point.*

**Response:** **Thanks to the reviewer for this comment. The title of the figure has been changed. Please see Fig. 3 caption in the revised manuscript.**

**"Fig 3: Correlation coefficient between (a) Eastern EA (yellow box; 30° E-50° E), (b) Congo Basin (yellow box; 15° E-30° E), and (c) Western EA (yellow box; 10° E-15° E) rainfall with SST during 1981-2023 period. The oceanic boxes are the same as those in Fig. 2. The stippling occurs where the correlation is statistically significant at the 95% confidence level through the Student's t test. The SST and rainfall data come from ERSST and CHIRPS, respectively."**

*The order in which things are discussed in the text is wrong. The sequence is 3c, 4, 3b, then 3a.*

**Response: This comment is taken into account. Please see the revised manuscript.**

*Line 290 The statement too strong. Although the anomalies are not significant, the dipole is clearly seen in the correlation patterns.*

**Response: Although a correlation dipole is present, these correlations are weak (|r|< 0.2), suggesting a very weak linear relationship between DMI and rainfall over this region. However, Moihamette et al (2024) showed that rainfall over central Africa along the Atlantic coast is significantly influenced by the Atlantic Ocean during IOD episodes, induced by its teleconnection with the Indian Ocean via atmospheric bridges and oceanic pathways. This indicates a weak linear relationship between precipitation in this region and the IOD.**

*Fig. 4 The caption is wrong: b and c are not "as in a" because "a" is precipitation. Also, is DMI the DMI index. The caption implies it is SST averages over the DMI region, but that make no sense. DMI is calculated from two regions which generally have opposite SST anomalies.*

**Response: Thank you for your comment. The title of the figure has been changed. See Figure 4 caption.**

**"Fig 4: (a) Indices of standardised rainfall anomalies over 1981-2023, averaged over the red box indicated in Fig. 1. (b) Indices of standardised SST anomalies over 1981-2023, average over the NTA and ATL oceanic regions. (c) As in (b), but for the DMI and ENSO index averaged over the IOD and Niño-3.4 oceanic regions. The SST data come from ERSST."**

*In the discussion of Fig. 6, the Hastenrath papers really need to be included.*

**Response: This comment is taken into account. Please see the revised manuscript in line 414.**

---

## Author Comment (AC2)

Dear Reviewer #2 Thank you for your careful review of our manuscript. Your comments are greatly appreciated and we think this new version of the manuscript responds to your concerns and provides an interesting contribution to the study of November 2023 extreme rainfall. The pdf file containing the response is attached.

**Anonymous Referee #2**

*The authors investigate atmospheric and ocean drivers of the Novermber 2023 extreme high rainfall month in equatorial Africa. They study a number of drivers previously studied in the literature in relation to the season, and place the 2023 event in the context of the climatology. The authors highlight in their introduction a number of direct and dramatic impacts arising from the extreme weather, and therefore justify the focus on this particular period.*

*The paper is well-written and provides a useful description of factors influencing an important extreme month of weather. The most persistent issue I had with this paper is the poor description of figure annotations. That is easily sorted, but I place the comment under "major" so the issue is clear. There are a few minor comments I have that either require small corrections or re-consideration by the authors. Once these are addressed, I can see the paper being publishable.*

*Major comments*
*Figs1, 3, 5,68, 9, 10 and 12 – all include boxes or lines annotating the figure, but which are not mentioned in the caption. Please explain what they are in the captions.*
**Response: Thanks to the reviewer for this comment. These comments are taken into account. Please see the caption of Figs 1, 3, 5,6, 8, 9, 10 and 12 in the revised manuscript.**

*Minor comments*
*Sec2.1 - says you use ERA5 SST and ERSST for sea surface temperatures. Be clear throughout which you are using. I suggest you add to the caption of every relevant figure, unless there is a simple blanket statement you can make in the methods.*
**Response: This comment is taken into account. Please see the revised manuscript in Figures 3 and 4 captions.**

*L189 – what is "CB cell"? It is not defined anywhere. Given it's referred to a number of times, it might be worth annotating on an early plot.*
**Response: This comment is taken into account. Please see the revised manuscript and the text adjusted. Lines 192-205**

*L195 – Can you explicitly state if this is mean to be a scalar (as it is in the equation you present)? Often I would think V might be vector of u and v winds, with Q also being a vector with u and v components (or even z component too). Are you using "wind speed" magnitude for V? I.e. you are losing the direction information?*

Response: **This comment is taken into account. Please see the revised manuscript. Lines 211-212**

*L201 – add the term "climatology" or similar to be clear about this. Also in the caption*

Response: **This comment is taken into account. Please see the revised manuscript. Line 217**

*L203 - "long-term mean (LTM) NOVEMBER rainfall" - "novermber" is needed.*

Response: **This comment is taken into account. Please see the revised manuscript. Line 219**

*L280-282 – You highlight the contradiction in the DMI term between the 2019 and 2023 response. However, I can't see that you explain why they are different. If you cannot explain it, can you be explicit at this point in the text and say so. If you do explain it, can you give a brief mention to what the discerning factor was at this point in the text.*

Response: **In this sentence, we want to show that, although the DMI in 2019 was slightly stronger than in 2023, precipitation was more heavy in 2023 than in 2019. This explains the occurrence of additional factors, such as the Niño-3.4, which was present in 2023 but absent in 2019. This comment is taken into account and the text adjusted. Please see the revised manuscript. Lines 284-285**

**"Notably, the DMI magnitude in 2023 was smaller than in both November 1997 and 2019, when conditions in EA were considerably drier than in 2023, suggesting that additional factors may have contributed, such as the Niño-3.4, which was absent in 2019 but present in 2023."**

*L334-334 - "These LLWs…" sentence. I'm not sure this sentence is entirely precise or correct. I'm not sure what you're saying is cooling (subsiding air warms as it follows the dry adiabat). Is this just about land ocean temperature contrasts, and consequent changes in circulation. It sounds like your suggesting the circulation is causing the colder temperature over the ocean. Can you reconsider this sentence, and ensure your confident in it? Ideally, provide a reference which justifies the statement.*

Response: **This comment is taken into account and the text has been rewritten. Please see the revised manuscript in lines 353-354.**

**"These LLWs are controlled by the heating contrast between land and ocean (Pokam et al. 2014)."**

*L375 - "...upper troposphere..." - Your plot only goes to 400hPa. I'd say 400hPa is borderline upper troposphere so I'm not sure it's a fairly persuasive statement. I think you'd either need to show the anomalies make it to 300-200hPa, or just say they do stretch deep into the troposphere up to 400hPa.*

**Response: This comment is taken into account. The figure has been redone and the text adjusted. Please see the revised manuscript, Figure 5 and line 394.**

*Fig 8 caption – "Omega" has units hPa s-1. Vertical velocity, w, has units m s-1. Please be clear about which is being plotted here.*

**Response: This comment is taken into account. The vertical velocity unit is Pa s$^{-1}$. The text has been adjusted. Please see the revised manuscript in the lines 152-153 and Figure 8 caption.**

*Fig8 cpation – what does the 10E-3 ms-1 refer to? Is it the value of the arrow on the figure panels? If so it must be the units of the horizontal winds too, not just vertical velocity? This needs clarity*

**Response: This comment is taken into account. $10^{-3}$ Pa s$^{-1}$ unit refers only to the vertical velocity (shade in Fig. 8). Please see the revised manuscript in the Figure 8 caption.**

*L477 - "no vectors have entered..." - I'm not sure what you mean here. Whether arrows actually cross the box edges is an arbitrary choice of plotting position for the arrows. There clear are some meridional components to the winds along those 2 boundaries. I suggest you need to be precise about what you mean and quantify it. Do you mean no net meridional component along the boundary relative to the size of the net zonal component at the east and west boundaries?*

**Response: This comment is taken into account and the text has been adjusted. As for the western and eastern borders of the EA, we note a weakness of the meridional component along the northern and southern borders compared to the size of the zonal component. Please see the revised manuscript in lines 498-500.**

**"As for the western and eastern borders of the EA, we note a weakness of the meridional component along the northern and southern borders compared to the size of the zonal component."**

*L517 – I'm not sure the red contours would be called "southwest". They look like "central" or "west central" to me, since the very southwest (10-20E, 6-10S) is white contours.*

**Response: This comment is taken into account and the text has been adjusted. Please see the revised manuscript in line 541.**

**"except in south-western CB and northern Angola (10°-20° E, 6°-10° S)"**

***Technical comments***

*L149 – kgkg-1 should have small letter "k"*
**Response: This comment is taken into account and the text has been adjusted. Please see the revised manuscript in line 152.**

*L345 - "JAS, The" needs to be a lower-case "t"*
**Response: This comment is taken into account and the text has been adjusted. Please see the revised manuscript in line 364.**

*L462 – "tropospheric" to "troposphere"?*
**Response: This comment is taken into account and the text has been adjusted. Please see the revised manuscript in line 483.**

*L506 - "confirmes" typo*
**Response: This comment is taken into account and the text has been adjusted. Please see the revised manuscript in line 529.**

*L557 - "favorasing" typo*
**Response: This comment is taken into account and the text has been adjusted. Please see the revised manuscript in line 581.**